# Predictors and Consequences of Not Seeking Healthcare during the COVID-19 Pandemic: Findings from the HEAF Cohort

**DOI:** 10.3390/ijerph192013271

**Published:** 2022-10-14

**Authors:** Stefania D’Angelo, Gregorio Bevilacqua, Ilse Bloom, Georgia Ntani, Karen Walker-Bone

**Affiliations:** 1MRC Lifecourse Epidemiology Centre, University of Southampton, Southampton SO15 3BX, UK; 2MRC versus Arthritis Centre for Musculoskeletal Health and Work, MRC Lifecourse Epidemiology Centre, University of Southampton, Southampton SO15 3BX, UK; 3Monash Centre for Occupational and Environmental Health, Monash University, Melbourne, VIC 3004, Australia

**Keywords:** COVID-19, healthcare utilisation, health seeking behaviour, middle-aged people, mental health, physical health

## Abstract

The COVID-19 pandemic resulted in a dramatic reduction of routine healthcare availability in many European countries. Among a cohort of English middle-aged adults, we explored pre-pandemic and pandemic factors associated with not seeking healthcare during lockdown, and their effect on subsequent self-reported health measures. Longitudinal data from the Health and Employment After Fifty (HEAF) cohort were used. Pre-pandemic data came from the 5th annual follow-up (2019), when participants were aged 56–71 years, and pandemic data were collected by e-survey in February 2021 and November 2021. Response rates of the two e-surveys were 53% and 79%, respectively. Pre-pandemic predictors of not seeking healthcare were: female gender, higher BMI, higher comorbidity, poorer self-rated health and depression; non-care seekers were also more likely to report that family or friends were affected by COVID-19 and to have been advised to shield. Not seeking healthcare during lockdown was associated with a higher risk of reporting worsening of physical, but not mental, health during the later phase of the pandemic. In this cohort, those with generally poorer health were disproportionately more likely to not seek healthcare during lockdown, which may potentially exacerbate pre-existing inequalities and lead to longer-term health consequences.

## 1. Introduction

In March 2020, the World Health Organisation (WHO) declared the infectious disease caused by SARS-CoV-2 a pandemic. To date, this virus has led to more than 500 million infections and caused the death of over 6 million people globally [1]. In order to reduce transmission of the disease and avoid overwhelming their healthcare systems, many countries adopted restrictive measures such as social distancing, lockdowns, curfews, and requirements to stay at home [2,3]. There is evidence from healthcare databases that access to routine and, in some cases, emergency healthcare throughout Europe was severely restricted [4,5,6]. Compared with pre-pandemic times, numbers of consultations in both primary care and emergency departments reduced substantially [4,5,6,7,8]. Data from Scotland showed that visits to Emergency departments dropped dramatically as soon as the pandemic was declared [9]. Likewise, UK data from electronic health records showed that consultations in primary care for key physical and mental health conditions decreased considerably compared with the pre-lockdown period [10]. As a consequence, fewer diagnoses of common health conditions such as diabetes, circulatory system disease and common mental health problems like depression, and anxiety were made [4,11]. In Italy, there have also been reports of a widespread fall in cancer diagnoses [12]. A systematic review has brought together evidence from studies looking at changes in healthcare utilisation from the pre- to the post-pandemic period and has concluded that healthcare utilisation decreased globally by about a third [8].

Availability of healthcare changed dramatically during the early pandemic: firstly, all available healthcare resources were mobilised to tackle the COVID-19 emergency, leading to postponement and cancellation of a great deal of planned, elective inpatient and outpatient care [13]. Secondly, the public was actively encouraged to only seek healthcare if they absolutely needed it, to avoid overloading healthcare systems. In some cases this led people who would normally have sought healthcare for their symptoms to voluntarily refrain from doing so, either because they were concerned about the risk of exposing themselves to COVID-19 or because they judged that their symptoms were not important or severe enough to require accessing healthcare [14]. Moreover, the pandemic led to a rapid expansion in telehealth to minimise the number of patient contacts with staff and each other. However, healthcare that is virtual can risk widening disparities among marginalised populations including older people [15]. Whatever the reason, reduction in availability and access of healthcare during the pandemic has left a legacy of delayed, or in more extreme cases, missed assessment of symptoms that in turn result in delayed diagnoses for conditions like diabetes, cardiovascular disease, mental health problems and cancer, which, when left untreated, may lead to potentially longer-term and major consequences for the health of individuals and the population [14,16].

In this study, we used longitudinal data from the Health and Employment After Fifty (HEAF) cohort to investigate: (1) what pre-pandemic factors predicted actively choosing not to seek healthcare during the first lockdown and (2) what were the consequences on subsequent self-reported health of not accessing healthcare when they would normally have done so.

## 2. Methods

The methods of recruitment of the HEAF cohort have been described in detail elsewhere [17]. Briefly, a population sample of adults then aged 50–64 years was incepted with the support of 24 English general practices. The practices sent study packs to all patients registered with them who were in the right age group in 2013–2014. Willing participants returned their baseline questionnaire to the study team along with written consent to be sent annual questionnaires thereafter. Each questionnaire covered demographic factors, markers of psychological and physical health, employment circumstances and retirement intentions. The most recent questionnaire follow-up was mailed in 2019.

In February 2021, all participants who had provided an e-mail address were contacted and asked whether they would contribute to a new sub-study by completing an online survey focussed on detecting changes to several aspects of their lives since the start of the COVID-19 pandemic. The survey was administered on the online platform Qualtrics (Provo, UT, USA), and covered the period of the first UK national lockdown (March–July 2020). Questions covered aspects such as personal experiences of COVID-19 and shielding, healthcare utilisation, employment circumstances before and after March 2020, finances, mental and physical health, social isolation, lifestyle factors and changes to people’s lives since the start of lockdown. Respondents were asked to confirm their consent for participation in this sub-study and were informed they could withdraw their consent at any time. Those who completed this first survey were asked to complete a second survey in November 2021, with the aim of understanding what further changes had occurred during the pandemic.

### 2.1. Outcomes

Participants were asked whether, since lockdown, they had not reported symptoms of an illness to a GP or other health professional when they usually would have done so. We defined them as “actively choosing not to seek healthcare”. Participants were only asked to tick this box if they believed that this applied to them, therefore all missing answers were recoded as “not experienced”. This outcome was assessed at the first timepoint (first online survey, February 2021).

Self-perceived worsening of mental and physical health since the beginning of lockdown was assessed at the second timepoint (second online survey, November 2021) by asking participants whether they agreed or not (5-level Likert scale) with the statement that their mental/physical health had deteriorated. Responses were recoded and analysed as “Agree” vs. “Neither/disagree”.

### 2.2. Predictors of Healthcare Avoidance

Factors that were tested as potential predictors of healthcare avoidance during lockdown were collected either with the most recent questionnaire pre-pandemic or with the first online survey. The online survey collected information about employment status in February 2020 (this was coded as “in work” vs. “not in work”) and changes in employment status since the start of lockdown (“already retired pre-lockdown”; “working in the same place”, “working from home”, “furloughed”, “not working—other reasons”). Participants were also asked whether their families or friends had been affected in some way by COVID-19, whether they were living alone during lockdown, and whether they had been advised to shield. Factors collected in 2019 included self-reported ability to manage financially (coded as “comfortably” “doing alright” vs. “just about or worse”); self-rated health (SRH) assessed with a single 5-scale question with options ranging from “excellent” to “poor” (and recoded as “at least good” vs. “fair/poor”); depressive symptoms (assessed by the Centre for Epidemiological Studies Depression [CES-D] [18] with scores ≥16 indicative of depressive symptoms) and body mass index (BMI). Finally, number of comorbidities reported in 2019 was computed and grouped as “none”, “1 to 3”, “4 or more”.

### 2.3. Statistical Analysis

Characteristics of participants were described, comparing those who reported that they had “actively chosen not to seek healthcare” and those who did not. Numbers and percentages were used for categorical variables and means (SD) for continuous normally distributed variables. Poisson regression model with the option for robust standard error was employed to explore the association between each risk factor and not seeking healthcare, with estimates expressed as relative risks (RRs) and 95% Confidence Intervals (95%CIs). After running univariate analyses, a fully adjusted model was built with all factors significant in the univariate analyses included in the same regression model.

Poisson regression with robust standard error was also used to explore the association between not seeking healthcare during the first lockdown and self-perceived worsening of health. Analyses were carried out with Stata software v17.0.

## 3. Results

### 3.1. Risk Factors for Not Seeking Healthcare

A total of 8134 men and women were originally recruited to the HEAF study. Of those, 4665 had returned a questionnaire in 2019 and provided us with a usable email address and were therefore invited to take part in the first COVID-19 online survey in February 2021. Of the participants invited, 44% were men, 38% were financially comfortable while 24% were struggling financially and 16% rated their health as fair or poor. In total, 2469 (53% response rate) completed the online survey and 289 (11.7%) participants reported that they had actively chosen not to seek healthcare for symptoms during the first lockdown. Table 1 shows that the prevalence of not seeking healthcare was higher among: women, those whose family or friends had been affected by COVID-19, and those struggling financially pre-pandemic. People with higher BMI, in fair/poor SRH, with depressive symptoms, reporting 4 or more comorbidities, and those advised to shield were also more likely to have not actively sought healthcare. No significant differences in healthcare seeking emerged for employment status pre-pandemic, changes in employment status during lockdown, or living alone in lockdown. In the mutually adjusted model including all significant risk factors from univariate analyses and age, women were at increased risk of not seeking healthcare (RR = 1.46; 95% CI = 1.16 to 1.84); as well as those with higher pre-pandemic BMI (Overweight vs. Normal RR = 1.41; 95% CI = 1.08 to 1.86 while Obese vs. Normal RR = 1.42; 95% CI = 1.05 to 1.92); and those reporting 4 or more comorbidities, depression and fair/poor SRH (RR = 1.41; 95% CI = 1.09 to 1.85). Among factors assessed during the pandemic, people with family/friends affected by COVID-19 remained at increased risk of not seeking healthcare even if needed (RR = 1.60; 95% CI = 1.28 to 1.98). The effects of the other factors were attenuated after full adjustment.

### 3.2. Not Seeking Healthcare and Health Worsening

To address the second research question, we restricted the analysis sample to participants (*n* = 1976) who returned a usable questionnaire at the second timepoint (November 2021). We also excluded 24 participants with missing data for both outcomes (worsening in mental and physical health) and therefore *n* = 1952 participants (57% women) remained in this analysis. In total, 17% of participants reported a worsening in mental health since the beginning of lockdown, with prevalence ranging from 34% among those who did not choose to seek healthcare at the first timepoint to 14% among those who did. Twenty-seven percent of the whole sample reported that their physical health worsened, and just over 50% of those who did not seek healthcare during the first lockdown reported such deterioration.

Participants who reported that they had not actively sought healthcare at the first timepoint (vs. those who did) were significantly more likely to report worsening of physical health since the start of the pandemic. This association was robust to adjustment for age and all other factors included in Table 2 (RR = 1.34, 95% CI 1.08 to 1.67). However, the same relationship was not seen with worsening of mental health.

## 4. Discussion

In this longitudinal study, we explored pre-pandemic and pandemic factors associated with not seeking healthcare during the first UK national lockdown in a cohort of middle-aged men and women living in England. We also explored whether not seeking healthcare in the first stages of the pandemic was associated with subsequent responses about their mental and physical health. In our sample, we found that women, those with family or friends affected by COVID-19, those advised to shield, with pre-pandemic higher BMI, higher number of comorbidities, fair/poor SRH and depression were more likely to have chosen not to seek healthcare during the lockdown. However, not seeking healthcare was not associated with either change in employment status or living alone during lockdown. After 8 months of follow-up during the pandemic, individuals who reported not seeking healthcare previously were more likely to report that their physical health (but not their mental health) had deteriorated.

Our findings of an increased risk of not seeking healthcare during lockdown among women and those in poorer health are in line with findings from previous cross-sectional studies [14,19,20]. It has been previously reported that, when dealing with health concerns, women have a higher perception of risk and tend to display more precautionary behaviour than men [21]. As per the health status, in these studies this variable was assessed in different manners: for example, Soares et al. [20] measured number of diseases and self-rated health, and reported that having any underlying disease (as compared with having none) was only mildly associated with the outcome, while participants rating their health as bad or very bad (as opposed to those rating it as good/very good) were 38% more likely to avoid healthcare. Czeisler et al. [19] showed that having any number of underlying medical conditions (as opposed to none) was significantly associated with reporting delay or avoidance of medical care, while a study by Splinter and colleagues [14] only explored self-appreciated health and found a 58% increased odds of healthcare avoidance for each unit decrease in the predictor. To our knowledge, an association between not seeking healthcare and overweight or obesity has not been reported elsewhere. As obesity is a common risk factor for comorbid diseases including type 2 diabetes, cancer and cardiovascular disease [22], this finding may well also indicate that markers of poorer general health were important in determining avoidance of healthcare seeking during lockdown.

Studies carried out pre-pandemic show consistently that people in generally poorer health are more likely to access healthcare compared with people with fewer healthcare needs [23,24]. Indeed, individuals with a greater number of health problems would understandably make more use of healthcare service, in the absence of a global health concern such as a pandemic. Therefore, our findings suggest a reversal of this (expected) trend occurred in the early phase of the pandemic so that those most in poorest general health and therefore most likely to need medical care avoided accessing that care. It may be that those with comorbidities and poorer health perceived themselves more likely to die from COVID-19 infection and therefore chose not to seek care and certainly there was publicity about an increased risk of death amongst older people and those living with obesity.

Our finding that depression was a factor associated with no-healthcare seeking was also reported in a cross-sectional study of US adults reported that during the pandemic, depressive symptoms in the previous week were associated with an increased risk of delay and avoidance of medical care [25]. Indeed, it is recognised that depression and mental distress are factors associated with healthcare avoidance even in in a pre-pandemic setting [26,27].

In our population sample, approximately 12% of participants voluntarily chose not to seek healthcare during the first UK national lockdown. A cross-sectional study of middle-aged Dutch residents conducted in the initial phase of the pandemic reported a comparable prevalence of non-healthcare seeking [14]. However, other studies have reported a much higher prevalence of healthcare avoidance. In a cross-sectional survey of 1000 responders with a mean (SD) age of 47.04 (15.04) years conducted in South Korea, an alarming 73% of participants reported that they avoided accessing hospitals during lockdown even when they were sick [28]. Such big disparity might be due to cultural differences between countries, differences in the measure of healthcare avoidance adopted, and/or be a reflection of different national guidance issued by the UK and South Korean Governments. A survey of adults aged 18 and older conducted in Portugal in July 2020 showed that 44% of the sample reported that they had either not scheduled or had postponed medical appointments or non-urgent treatments [20], and similarly an estimated 41% of US adults reported to have delayed or avoided medical care during the first phase of the pandemic because of concerns about COVID-19 [19]. One of the reasons for such stark differences in the prevalence of the outcome might be due to the definition of healthcare avoidance employed, which in the case of the HEAF study, only includes avoidance of healthcare and not postponement; or could also reflect differences in healthcare systems between countries. Importantly, participants of the HEAF study are significantly older than the participants in some of these other pandemic surveys which may also be a factor. An implication of this is that more healthcare need would be anticipated amongst this older demographic.

In our study, the only pandemic-related factor that was associated with not seeking healthcare was reporting having family or friends affected by COVID-19. It could be hypothesised that this was caused by a heightened awareness of the possible consequences of contracting the virus. However, this association was not found in a study by Czeisler et al. [19], who found that neither “knowing someone who tested positive for SARS-CoV-2” nor “knowing someone who died from COVID-19” had affected healthcare seeking during the pandemic. This inconsistency between our study and Czeisler’s can be ascribed to the different way they assessed the variable of reporting family or friends being affected by COVID-19: Czeilsler et al. tested separate risk factors such as “knowing someone who tested positive for SARS-CoV-2” and “knowing someone who died from COVID-19” [19], whereas we asked participants about any sort of impact the pandemic might have had on their family or friends.

Our study is novel in quantifying the potential health consequences of not seeking healthcare in a global pandemic. Our hypothesis that underusage of healthcare might lead to worsening of health appeared to be borne out, at least in relation to short-term physical health. It will be an important next step to evaluate these effects over the longer term and their implications for healthcare provision. This study covers the period up to November 2021 and further research is needed to explore the long-term indirect effect of the COVID-19 lockdown on people’s health and health-related behaviours.

Our findings need to be considered alongside some limitations: firstly, we have previously shown [29] that this survey suffers from a responder bias which limits the generalisability of the findings, as participants who returned a usable questionnaire reported better socio-economic position and health status at baseline compared with the remaining consenting participants. Based on our findings of increased risk of not seeking healthcare among people with poorer health, this could mean that the prevalence of not seeking healthcare might be underestimated in our study. Secondly, exposures and outcomes are all self-reported meaning that we could only observe a subjectively perceived health deterioration and not an objectively measured one. Finally, it might be that we are slightly overestimating the prevalence of not accessing healthcare as the variable “choosing not to seek healthcare” may include two distinct groups of people: those who have actively chosen not to access healthcare if they needed it or alternatively, those who did not need to access it in the absence of health concerns. On the other hand, to the best of our knowledge, this study is unique in its longitudinal design. The existing cohort allowed us to explore the effect of risk factors that pre-empted the outcome.

In summary, amongst a sample of adults aged 56–71 years who chose to avoid healthcare during lockdown, there were some subjective consequences in the short-term for their self-reported physical health. It remains to be seen if this translates into longer-term impacts.

## 5. Conclusions

Our findings suggest that those people already in poorer health pre-pandemic were disproportionately more likely to actively choose not to access healthcare during the pandemic. Unsurprisingly, non-healthcare seeking was associated with poorer physical health at follow-up later in lockdown. These findings could have long-term implications for the health of individuals who were already disadvantaged pre-pandemic.

## Figures and Tables

**Table 1 ijerph-19-13271-t001:** Risk factors for not seeking healthcare.

	Healthcare Use During First Lockdown				
	Seeking (*n* = 2180)	Not Seeking (*n* = 289)	*p*-Value	Univariate RR (95% CI)	Multivariate RR (95% CI)
Factors assessed within the COVID-19 online survey					
Sex					
Men	989 (90.9)	99 (9.1)	<0.001	Ref	Ref
Women	1191 (86.2)	190 (13.8)		1.51 (1.20, 1.90)	1.46 (1.16, 1.84)
Age, mean (SD)	65.6 (4.3)	65.5 (4.3)	0.58	0.99 (0.97, 1.02)	1.00 (0.98, 1.03)
Employment status February 2020 (13 missing)					
Not in work	1308 (87.6)	186 (12.5)	0.18	1.16 (0.93, 1.46)	
In work	859 (89.3)	103 (10.7)		Ref	
Changes in employment in lockdown * (94 missing)					
Already retired pre-lockdown	1100 (88.7)	140 (11.3)	0.60	1.10 (0.80, 1.50)	
Working same place	410 (89.7)	47 (10.3)		Ref	
Working from home	386 (87.9)	53 (12.1)		1.17 (0.81, 1.70)	
Furloughed	91 (86.7)	14 (13.3)		1.30 (0.74, 2.26)	
Not working-others	114 (85.1)	20 (14.9)		1.45 (0.89, 2.36)	
Has your family been affected (38 missing)					
No	1267 (91.0)	126 (9.0)	<0.001	Ref	Ref
Yes	882 (85.0)	156 (15.0)		1.66 (1.33, 2.07)	1.60 (1.28, 1.98)
Did you live alone during lockdown (73 missing)					
No	1761 (88.7)	225 (11.3)	0.17		
Yes	352 (85.9)	58 (14.2)		1.25 (0.95, 1.63)	
Have you been advised to shield (4 missing)					
No	2002 (88.9)	250 (11.1)	0.006	Ref	Ref
Yes	174 (81.7)	39 (18.3)		1.65 (1.21, 2.24)	1.16 (0.85, 1.58)
Factors assessed within follow-up 5 in 2019					
Managing financially pre-pandemic (4 missing)					
Comfortably	1194 (91.1)	117 (8.9)	<0.001	Ref	Ref
Doing alright	699 (86.9)	105 (13.1)		1.46 (1.14, 1.88)	1.13 (0.88, 1.44)
Just about or worse	285 (81.4)	65 (18.6)		2.08 (1.57, 2.75)	1.19 (0.88, 1.62)
BMI (45 missing)					
Underweight	22 (84.6)	4 (15.4)	<0.001	1.89 (0.75, 4.77)	1.77 (0.71, 4.41)
Normal	856 (91.9)	76 (8.2)		Ref	Ref
Overweight	817 (87.6)	116 (12.4)		1.52 (1.16, 2.01)	1.41 (1.08, 1.86)
Obese	447 (83.9)	86 (16.1)		1.98 (1.48, 2.64)	1.42 (1.05, 1.92)
N. comorbidities (0 missing)					
0	896 (94.9)	22 (5.1)	<0.001	Ref	Ref
1 to 3	1224 (90.5)	128 (9.5)		1.86 (1.20, 2.88)	1.57 (1.01, 2.46)
4 or more	546 (79.7)	139 (20.3)		3.98 (2.58, 6.15)	2.38 (1.50, 3.79)
Depressed (CESD ≥ 16) (5 missing)					
No	1826 (90.6)	190 (9.4)	<0.001	Ref	Ref
Yes	349 (77.9)	99 (22.1)		2.35 (1.88, 2.92)	1.41 (1.09, 1.80)
Self-rated health pre-pandemic (10 missing)					
At least good	1885 (90.4)	200 (9.6)	<0.001	Ref	Ref
Fair/poor	286 (76.5)	88 (23.5)		2.45 (1.96, 3.07)	1.41 (1.09, 1.85)

* Those include people with missing data and people who were already unemployed before lockdown.

**Table 2 ijerph-19-13271-t002:** Characteristics of participants with mental or physical health worsening since the beginning of lockdown.

	Worsening of Mental Health *N* (%) *	Worsening of Physical Health *N* (%) *
Overall	327 (16.8)	525 (26.9)
Sex		
Men	95 (11.2)	205 (24.2)
Women	232 (21.0)	30 (28.9)
Advised to shield		
No	285 (15.9)	449 (25.0)
Yes	42 (26.6)	76 (48.1)
Healthcare use during the first lockdown		
Seeking	247 (14.4)	404 (23.5)
Not seeking	80 (34.3)	121 (51.9)
Managing financially in 2019		
Comfortably	134 (12.9)	207 (19.9)
Doing alright	119 (18.6)	201 (31.4)
Just about or worse	74 (27.7)	116 (43.5)
BMI in 2019		
Underweight	3 (13.0)	6 (26.1)
Normal	104 (13.8)	134 (17.8)
Overweight	135 (18.4)	206 (28.0)
Obese	81 (20.0)	170 (42.0)
N. comorbidities in 2019		
0	27 (7.8)	31 (9.0)
1 to 3	149 (13.9)	234 (21.8)
4 or more	151 (28.2)	260 (48.6)
Depressed (CESD ≥ 16) in 2019		
No	184 (11.4)	350 (21.8)
Yes	143 (42.3)	173 (51.2)
Self-rated health in 2019		
At least good	233 (14.1)	345 (20.8)
Fair/poor	92 (32.1)	177 (61.7)

* Missing data: worsening of mental health = 54; worsening of physical health = 11.

## Data Availability

The datasets used for this analysis are available on reasonable request from the MRC Versus Arthritis Centre for Musculoskeletal Health and Work by contacting the corresponding author.

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
