# Peer review of "Predictors and Consequences of Not Seeking Healthcare during the COVID-19 Pandemic: Findings from the HEAF Cohort"

_ijerph, 2022, doi:10.3390/ijerph192013271_

Round 1
Reviewer 1 Report
Dear Authors,
I've read your manuscript with interest.
It raises important issues related to health care and the need for it during the covid19 pandemic.
I think that your manuscript is an example of a well-planned and conducted study. It is a very innovative approach to the issue.
I like the large study sample and the research method used.
I understand that apart from the assessment of depressive symptoms, the other questions were of your own design. Did you estimate the internal consistency of the questionnaire before the study?
As GPs, do you intend to use the results in any way? Will you be repeating the study in some time?
Author Response
Reviewer 1
Dear Authors,
I've read your manuscript with interest.
It raises important issues related to health care and the need for it during the covid19 pandemic.
I think that your manuscript is an example of a well-planned and conducted study. It is a very innovative approach to the issue.
I like the large study sample and the research method used.
I understand that apart from the assessment of depressive symptoms, the other questions were of your own design.
Thank you for your very positive feedback about this manuscript. You are correct, the only validated tool that we used was to assess depressive symptoms. While the self-perceived worsening of mental and physical health was a bespoke measure.
Did you estimate the internal consistency of the questionnaire before the study?
Many thanks for your comment. We have not measured internal consistency because variables within the questionnaire were designed to measure different things and we felt that internal consistency would not be appropriate.
As GPs, do you intend to use the results in any way? Will you be repeating the study in some time?
Thank you for your comment. Our final aim is indeed to inform public health policies. We find it is particularly concerning to see that it was people in poorer health pre-pandemic that actively chose not to access healthcare during lockdown. Therefore, we hope by publishing this paper to identify the group most affected with the potential to explore any long-term health consequences. The HEAF study is ongoing, therefore as the study progresses, we will be in a position to explore the long-term health effects of not accessing healthcare in lockdown.
Reviewer 2 Report
Dear authors, thank you for giving me the opportunity to review your manuscript: “Predictors and consequences of not seeking healthcare during the COVID-19 pandemic: findings from the HEAF cohort”.
I think that this manuscript can contribute to the literature.
I send below some comments to improve the manuscript.
Introduction:
- Authors state that: “Whatever the reason, reduction in availability and access of healthcare during the pandemic has left a legacy of delayed diagnoses and treatments and has led to potentially longer-term consequences for the health of individuals and the population [15, 17].”
o Please, indicate and develop the consequences.
Methods:
- Please, explain better the assessment of depression: “depressive symptoms (assessed by the Centre for Epidemiological Studies Depression [CES-D] with scores ≥16 indicative of depressive symptoms)”.
- “…4,655 had returned a questionnaire in 2019…” – Please, provide more information about the sample (e.g., number of women and men, and other variables).
Discussion
- In some cases, the authors only indicate that other studies do or do not corroborate the results presented. It is also essential to explain those results. For example:
o “Studies carried out pre-pandemic show consistently that people in generally poorer health are more likely to access healthcare compared with people with fewer healthcare needs [22, 23].” – Please explain, according to the literature, why this is happening.
o “Our findings of an increased risk of not seeking healthcare during lockdown among women and those in poorer health are in line with findings from previous cross-sectional studies [15, 19, 20].” – Please explain, according to the literature, why this is happening.
o “In our study, the only pandemic-related factor that was associated with not seeking healthcare was reporting having family or friends affected by COVID-19. It could be hypothesised that this was caused by a heightened awareness of the possible consequences of contracting the virus. However, this association was not found in a study by Czeisler et al. [19], who found that neither “knowing someone who tested positive for SARS-CoV-2” nor “knowing someone who died from COVID-19” had affected healthcare seeking during the pandemic.” – Please, explain why these differences in results can arise.
- Authors should add implications for practice.
Author Response
Comments and Suggestions for Authors
Dear authors, thank you for giving me the opportunity to review your manuscript: “Predictors and consequences of not seeking healthcare during the COVID-19 pandemic: findings from the HEAF cohort”.
I think that this manuscript can contribute to the literature.
I send below some comments to improve the manuscript.
Thank you for your positive feedback about the manuscript and for your helpful comments.
Introduction:
- Authors state that: “Whatever the reason, reduction in availability and access of healthcare during the pandemic has left a legacy of delayed diagnoses and treatments and has led to potentially longer-term consequences for the health of individuals and the population [15, 17].”
o Please, indicate and develop the consequences.
Thank you for your comment. Several countries included the UK are experiencing a huge backlog of healthcare after the COVID-19 lockdowns. This of course translates to delayed, or in more extreme cases missed assessment of symptoms that in turn result in delayed diagnosis of conditions like diabetes, cardiovascular conditions, mental health problems which, when left untreated, may lead to major health consequences. In a country where health inequalities are well documented, our findings show that unless the unmet healthcare need for those in poorer health are tackled quickly, those inequalities might be exacerbated in the long term.
We have now changed the relevant sentence in our introduction (page 2, lines 60-64).
Methods:
- Please, explain better the assessment of depression: “depressive symptoms (assessed by the Centre for Epidemiological Studies Depression [CES-D] with scores ≥16 indicative of depressive symptoms)”.
Thanks for the opportunity to clarify. At baseline and each follow-up, we measured depressive symptoms using the Centre for Epidemiological Studies Depression (CES-D). This is a 20-item scale where participants were asked to report how often, in the previous 7 days, they experienced symptoms associated with depression, like poor sleep, loneliness, fearfulness. Possible options range from “Rarely or none of the time (less than once day)” to “Most or all of the time (5-7 days)”. A score is then computed by adding up all items, and this ranges from 0 to 60 (higher scores indicate higher depression). We’ve used a threshold of 16 or more to indicate presence of depression, as recommended in the following paper
Radloff, L. S. (1977). The CES-D scale: A self report depression scale for research in the general population. Applied Psychological Measurements, 1, 385-401.
In the interest of word count we have decided not to add the above description of the tool, but we have referenced the paper by Radloff
- “…4,655 had returned a questionnaire in 2019…” – Please, provide more information about the sample (e.g., number of women and men, and other variables).
We apologise for leaving out important information. There was also a typo in the original version of the manuscript as the number of people invited was 4,665 and not 4,655. Of those invited, 44% were men, 38% were comfortable financially while 24% were struggling financially, 58% owned their place outright. In terms of lifestyle, 7% were current smokers, while 60% were either overweight or obese. 16% rated their health as fair or poor.
We have now added this information (page 3, lines 137-138)
Discussion
- In some cases, the authors only indicate that other studies do or do not corroborate the results presented. It is also essential to explain those results. For example:
o “Studies carried out pre-pandemic show consistently that people in generally poorer health are more likely to access healthcare compared with people with fewer healthcare needs [22, 23].” – Please explain, according to the literature, why this is happening.
Thank you for this comment. Literature is consistent in finding that, although socio-economic status, age and gender are important determinants of accessing healthcare, self-perceived health status is the most important factor associated with healthcare seeking behaviour. People in poorer health might perceive themselves at higher risk and this translates into a higher usage of healthcare services. We added a brief sentence to the paragraph (page 6, lines 212-214).
o “Our findings of an increased risk of not seeking healthcare during lockdown among women and those in poorer health are in line with findings from previous cross-sectional studies [15, 19, 20].” – Please explain, according to the literature, why this is happening.
Thank you for this comment. Pre-pandemic literature consistently show that women usually seek healthcare more often than men (this is documented in the paper by Thompson et al, reference 23), therefore it’s interesting to see that during lockdown things changed and women became less likely to access healthcare. It is indeed a possibility that they might have felt more vulnerable during lockdown and had therefore made a conscious decision not to access healthcare. We have referenced an additional paper by Ibuka et al. which explains that women tend to have a higher perception of risk compared to men in regard to dealing with health concerns. (Additional text has been added on page 6, lines 193-195)
Regarding the point made about people in poorer health, it has been previously shown that people in poorer health chose not to seek healthcare during lockdown. This differs to the pre-pandemic period where those with poorer health were more likely to access healthcare. It is likely that people with poorer health status felt particularly vulnerable and at risk of catching the virus and therefore stayed away from healthcare.
o “In our study, the only pandemic-related factor that was associated with not seeking healthcare was reporting having family or friends affected by COVID-19. It could be hypothesised that this was caused by a heightened awareness of the possible consequences of contracting the virus. However, this association was not found in a study by Czeisler et al. [19], who found that neither “knowing someone who tested positive for SARS-CoV-2” nor “knowing someone who died from COVID-19” had affected healthcare seeking during the pandemic.” – Please, explain why these differences in results can arise.
Thanks for giving us the chance to expand on this. One possible explanation for such difference might be the variable used in our study and in the study by Czeisler and colleagues. While our definition included any kind of impact that the pandemic might have had on participants’ family (either testing positive, being hospitalised, or even died), Czeisler et al tested separately risk factors like “knowing someone who tested positive for SARS-CoV-2” and “knowing someone who died from COVID-19”.
We have added a sentence about this on page 7, lines 255-260
- Authors should add implications for practice.
Thank you for your comment, we have added the following conclusive sentence on page 8, lines 288-293
Our findings suggest that those people already in poorer health pre-pandemic were disproportionately more likely to actively choose not to access healthcare during the pandemic. Unsurprisingly, non-healthcare seeking was associated with poorer physical health at follow-up later in lockdown. These findings could have long-term implications for the health of individuals who were already disadvantaged pre-pandemic.